# The influence of cortical activity on perception depends on behavioral state and sensory context

Lloyd E. Russell [1], Mehmet Fişek[1], Zidan Yang[1], Lynn Pei Tan[1], Adam M. Packer [1], Henry W. P. Dalgleish[1], Selmaan N. Chettih[2], Christopher D. Harvey[2] & Michael Häusser [1] ✉

The mechanistic link between neural circuit activity and behavior remains unclear. While manipulating cortical activity can bias certain behaviors and elicit artificial percepts, some tasks can still be solved when cortex is silenced or removed. Here, mice were trained to perform a visual detection task during which we selectively targeted groups of visually responsive and co-tuned neurons in L2/3 of primary visual cortex (V1) for two-photon photostimulation. The influence of photostimulation was conditional on two key factors: the behavioral state of the animal and the contrast of the visual stimulus. The detection of low-contrast stimuli was enhanced by photostimulation, while the detection of high-contrast stimuli was suppressed, but crucially, only when mice were highly engaged in the task. When mice were less engaged, our manipulations of cortical activity had no effect on behavior. The behavioral changes were linked to specific changes in neuronal activity. The responses of non-photostimulated neurons in the local network were also conditional on two factors: their functional similarity to the photostimulated neurons and the contrast of the visual stimulus. Functionally similar neurons were increasingly suppressed by photostimulation with increasing visual stimulus contrast, correlating with the change in behavior. Our results show that the influence of cortical activity on perception is not fixed, but dynamically and contextually modulated by behavioral state, ongoing activity and the routing of information through specific circuits.

The perception of a sensory stimulus is modulated by the behavioral state in which it is experienced – successful detection of a stimulus can be increased during periods of arousal or attention[1,2]. The neural representation of the same sensory stimulus is also modulated by behavioral state[3–5] – stimulus-evoked responses are typically enhanced when subjects are alert. The behavioral state of an animal is a latent variable and can be inferred in a number of ways. Firstly, the arousal of an animal is thought to be under control of neuromodulatory activity

primarily arising from the locus coeruleus[6,7], which can influence the dilation and constriction of the pupil[8], with high mental effort and arousal associated with larger pupil sizes[9]. A second manifestation of behavioral state is the synchronization of neural activity[3,10]. At opposite extremes, sleep is associated with synchronized activity, while wakefulness is associated with desynchronization. The modulation of cortical responses by behavioral state[11–14], task outcome[15], and task demands[16,17] has been extensively investigated. However, how the

[1]Wolfson Institute for Biomedical Research, University College London, London, UK. [2]Department of Neurobiology, Harvard Medical School, Boston, MA, USA. ✉e-mail: m.hausser@ucl.ac.uk

modulation of cortical activity by state or stimulus corresponds to the influence of the cortex on behavior has largely been studied using only correlational methods[18,19]. While the correlational approach has yielded important insights, the absence of cellular-level manipulations of the activity patterns means a cause-and-effect relationship between the observations cannot be established.

Classical experiments have shown that electrically stimulating specific cortical areas can bias sensory perception[20–26] and elicit artificial percepts[27–30]. Optogenetic stimulation of cortex has confirmed and extended these findings[31–33]. In all of these experiments the functional identity of activated neurons was largely unknown, as was the number of activated cells. With the advent of new techniques that enable activation of a known number of functionally characterized cells, a view is emerging that perception can be initiated, or biased by, a small number of specific neurons[34–40]. However, challenging even the basic requirement of cortical neurons to solve some tasks, reversible silencing[41–45] or permanent lesioning[46–49] of cortex have produced contradictory findings about the necessity of cortical activity for perception and behavior[50–52].

Consequently, we lack a clear mechanistic link between cortical activity, how it engages local and downstream circuits, and ultimately how and when that activity influences a behavior of interest. To reconcile these disparate results and begin building a complete picture of how and when cortical activity leads to perception, we need to investigate the perceptual influence of specific patterns of neural activity during the processing of different stimuli in a variety of behavioral tasks and states.

To probe the influence of stimulus-relevant patterns of cortical activity on local network activity and behavior, we performed two-photon population calcium imaging of a volume of L2/3 V1 while simultaneously using two-photon holographic optogenetics[53–57] to activate specific groups of neurons in mice performing a visual contrast-varying detection task. We targeted groups of neurons based on their tuning for visual stimuli to ask whether increased neural activity in relevant neuronal ensembles leads to increased behavioral detection. This in vivo all-optical approach[58–67] also allowed us to assess the functional influence of the stimulated cells on the local network. Importantly, behavioral state was continuously monitored by measuring pupil size and neuronal synchrony and was used to index each behavioral session into states of higher and lower engagement, allowing us to investigate the impact of additional cortical activity during these different behavioral states. We found that photostimulation of task-relevant cells – defined as cells that preferentially respond to the orientation of visual stimulus selected for the detection task – impacted the non-photostimulated local network in a way that depended on the functional identity of the cells and the contrast of the visual stimulus. When the visual stimulus contrast was high, responses of neurons functionally similar to the photostimulated ensemble were more strongly suppressed than other cells. When the contrast was low these neurons were suppressed less than other cells. The effect of photostimulation on the animal's behavioral report followed a similar pattern, whereby the detection of high contrast stimuli was suppressed, but the detection of low contrast stimuli was enhanced. This behavioral effect, and the linking of cortical activity to behavior, was only present when mice were most engaged in the task. A gradual decrease in task engagement transitioned mice from a state where additional cortical activity bidirectionally influenced their behavior to one where such activity had no reliable impact.

## Results

### Behavioral state and task performance

To allow all-optical interrogation in visual cortex of mice performing a visually guided behavior, we co-expressed the calcium sensor GCaMP6s[68–70] with the excitatory, somatically-restricted opsin C1V1[71,72] in pyramidal cells of L2/3 V1. Mice were head-fixed and trained to perform a visual stimulus detection task (Fig. 1a, b) where after withholding licks for a random interval, a water reward could be obtained for successfully licking to report the appearance of a small drifting grating patch of randomized orientation (Fig. 1c, d). Mice learned the task quickly with a maximal contrast stimulus, reaching a high level of stable performance within days (Supplementary Fig. 1). Lowering the stimulus contrast reduced performance on the task (Fig. 1e, Supplementary Fig. 1). To assess the relationship between cortical activity, behavior and behavioral state, we recorded the animal's pupil size and neuronal synchrony throughout the behavioral session as a measure of arousal or alertness[13,73,74]. We observed a significant correlation between the size of the pupil and the degree of neuronal synchrony (Supplementary Fig. 2). We found that on threshold-stimulus trials (the stimulus contrast at which animals detected the stimulus on approximately half the trials), successful detections (hits) were associated with a more dilated pupil and lower neural synchrony in the period before the stimulus was presented (Fig. 1f, g; pupil diameter (normalized relative to the median of the session) on hits: $+2.5 \pm 10.8\%$ vs misses: $-6.6 \pm 13.7\%$, $P = 0.006$ Wilcoxon signed-rank test; neuronal synchrony (average pairwise Pearson's correlation coefficient) on hits: $0.0014 \pm 0.0009$ vs misses: $0.0024 \pm 0.0021$, $P < 0.001$, Wilcoxon signed-rank test). Based on this, we defined two behavioral states: one associated with a large pupil size and low neuronal synchrony, and the other with a smaller pupil size and higher neuronal synchrony (Fig. 1h). We split each session into these two states, indexing each trial by the pupil size and neuronal synchrony in the period before the stimulus appeared, and then compared the resulting psychometric curves between the two states (Fig. 1i). We found that performance was higher in the state with larger pupil size and lower neuronal synchrony – corresponding to a more-engaged state – consistent with previous reports[4,75–77] (Fig. 1i, d-prime averaged across all contrasts in the more-engaged state vs the less-engaged state: $1.52 \pm 1.13$ vs $1.32 \pm 1.05$, $P = 0.0002$ Wilcoxon signed-rank test). The more engaged state was also characterized by a greater detection sensitivity (Fig. 1j, width of psychometric curve in the more-engaged state vs the less-engaged state: $7.5 \pm 8.3$ vs $15.0 \pm 15.2$, $P = 0.012$ Wilcoxon signed-rank test) and lower stimulus contrast threshold (Fig. 1j, threshold of psychometric function in the more-engaged state vs the less-engaged state: $3.88 \pm 1.47$ vs $4.59 \pm 2.11$, $P = 0.042$ Wilcoxon signed-rank test) reflecting greater arousal or engagement in the task[75,78,79]. Importantly, we note that the less-engaged state does not correspond to a completely disengaged state, as mice were still performing the task to a high level. This subtle but clear distinction provides the opportunity to quantitatively compare conditions under which cortical activity may or may not play a role in shaping simple behaviors.

### Behavioral effects of targeted photostimulation

To test the influence of activity in a stimulus-encoding population of neurons in V1 during different behavioral states, we targeted multiple cells for two-photon optogenetic photostimulation while recording the resulting neuronal and behavioral changes. We first identified the retinotopically appropriate and co-expressing field of view and then mapped the visual stimulus-responsive and photostimulation-responsive neurons (Fig. 2a, b). We then selected the maximum number of photostimulation-responsive neurons that shared the same visual stimulus orientation preference (median ~20 neurons, Fig. 2c–e). The orientation of the visual stimulus for the detection task was chosen to match the orientation preference of this neuronal ensemble. While the mice were performing the contrast-varying detection task, we photostimulated these co-tuned ensembles on a random subset of trials with and without concurrent visual stimulus presentation (Figs. 1c, d and 2f). Intriguingly, we observed a reliable effect of photostimulation on behavior only when the animal was more engaged. In this large pupil, low synchrony state, targeted photostimulation of ~20 neurons enhanced the detection of lower contrast visual stimuli, while

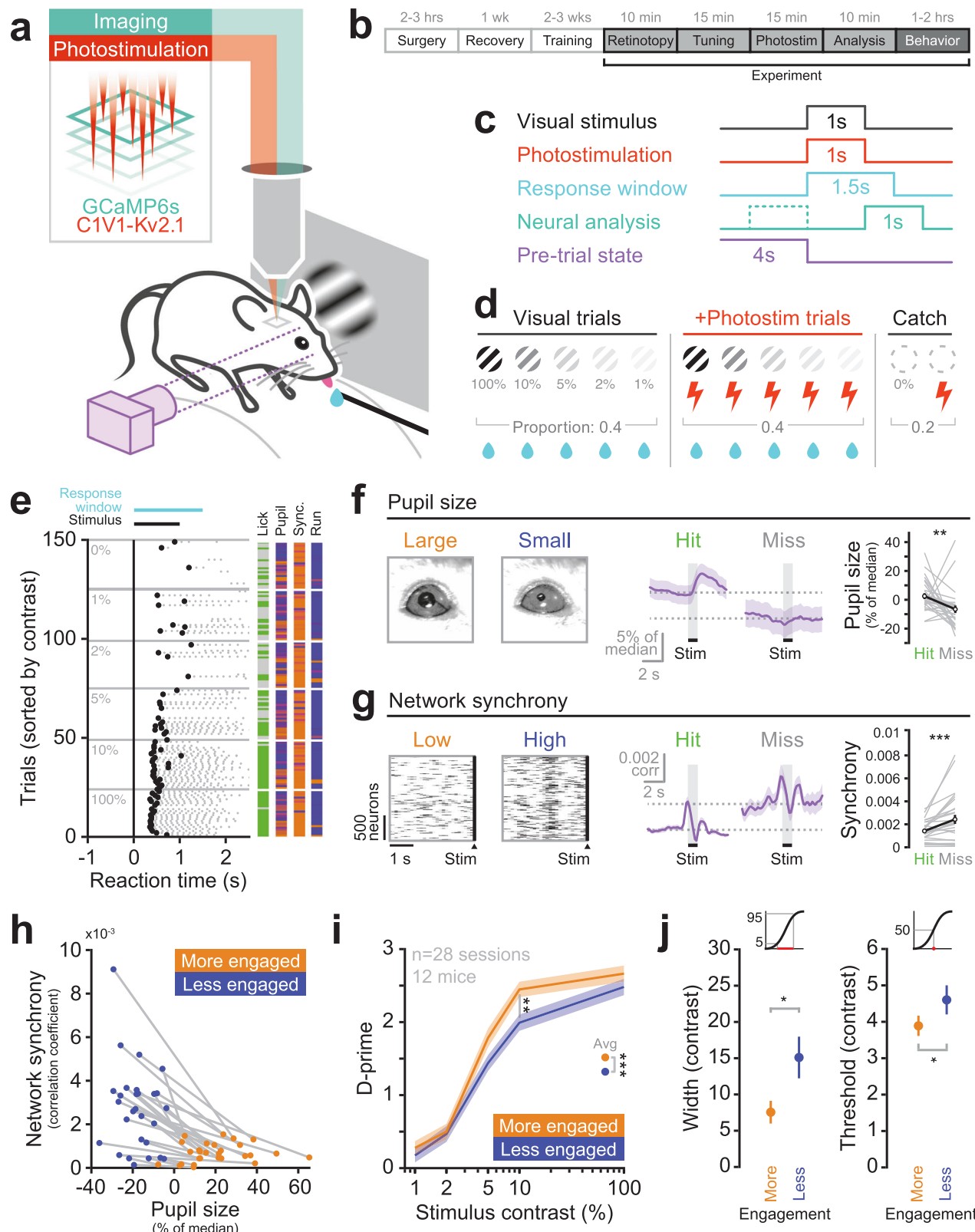

the same photostimulus suppressed the detection of higher contrast visual stimuli (Fig. 2f, h; width of psychometric curve in the less engaged state without vs with photostimulation: $15.0 \pm 15.2$ vs $13.3 \pm 32.5$, $P = 0.130$ Wilcoxon signed-rank test. Width of psychometric curve in the more engaged state without vs with photostimulation: $7.5 \pm 8.3$ vs $25.4 \pm 51.6$, $P = 0.0015$ Wilcoxon signed-rank test. The more-engaged state vs the less-engaged state, repeated measures ANOVA,

interaction of contrast and state $P = 0.009$). Taken together, photo-stimulation acted to widen the psychometric function, reducing sensitivity[80] (Fig. 2g; change in width in the more-engaged state vs the less-engaged state: $17.9 \pm 49.2$ vs $-1.8 \pm 35.8$, $P = 0.036$ Wilcoxon signed-rank test) without changing the threshold (change in threshold in the more-engaged state vs the less-engaged state: $-0.2 \pm 1.6$ vs $-0.4 \pm 2.4$, $P = 0.982$ Wilcoxon signed-rank test). In other words,

**Fig. 1 | Visual stimulus detection is modulated by behavioral state. a** Experiment schematic. Mice coexpress GCaMP6s and C1V1-Kv2.1 in excitatory neurons of L2/3 V1 enabling simultaneous two-photon calcium imaging and two-photon holographic stimulation. Mice are head-fixed and trained to perform a visual stimulus detection task. Pupil size is recorded with a camera. **b** Timeline of animal preparation, training, and experiment. **c** Behavioral trial structure. After withholding licks for a randomized interval (4 ± 3 s) a stimulus (visual and/or optogenetic) is presented to the mouse. When a visual stimulus is presented the mouse has 1.5 s to lick the lickometer in order to receive a water reward. Neural analysis is performed in the 1 s immediately following stimulus offset (to avoid photostimulation artifact) using the 1 s immediately prior to the stimulus as baseline. The state of the animal is measured in the 4 s preceding the delivery of the stimulus. **d** Behavioral session structure. 12 different trial types are presented to the mouse in a pseudorandom blocked structure. Visual-only trials (40% of session) of varying contrast (1%, 2%, 5%, 10% and 100%) are interleaved with Visual+Photostimulation trials (40% of session) where a 1 s 20 Hz photostimulation is delivered coincident with the visual stimulus. Catch trials (20% of session) with and without photostimulation are delivered to assess chance licking probabilities. Any trial with a visual stimulus is rewarded if the mouse responds during the response window. **e** Example lick raster plot. Trials were delivered pseudorandomized but are shown sorted by stimulus contrast. Gray dots indicate lick responses, with the first lick (reaction time) highlighted in black. Right: the simultaneously recorded pre-trial pupil size, neuronal synchrony and running speed are shown for each trial. Orange indicates large values, blue indicates small values. **f** Pupillometry is performed throughout the behavioral session. Large and small pupil sizes are seen (left) reflecting different behavioral states. Stimulus triggered average pupil traces are averaged across all hit or miss trials of threshold stimuli, and then averaged across sessions (middle). On visual-only threshold trials

hits (mice licked within response window) are associated with a larger pupil prior to the stimulus delivery (right). Error bars show mean ± SEM across sessions, $n = 28$ sessions, 12 mice. **g** Neuronal synchrony in the pre-trial period is computed as the average pairwise correlation between all pairs of cells' deconvolved activity traces. Periods of low and high synchrony are seen (left) reflecting different behavioral states. Stimulus triggered average correlation traces are averaged across all hit or miss threshold trials, and then averaged across sessions. Note the traces are made with a 4 s sliding window hence the synchrony appears to increase before the stimulus is delivered (middle). On threshold contrast visual-only trials hits are associated with a lower level of synchrony prior to the stimulus delivery (right). Error bars show mean ± SEM across sessions, $n = 28$ sessions, 12 mice. **h** Each session is median-split into two collections of interspersed trials, based on pupil size (normalized relative to the median size in the session) and network synchrony (average pairwise Pearson's correlation coefficient between all recorded neurons) on each trial preceding stimulus delivery. The more engaged state contains trials with the largest pupil sizes and lowest neuronal synchrony. The less engaged state contains trials with the smallest pupil sizes and highest neuronal synchrony. Each dot represents the average pupil size and synchrony for that state of a given session. Gray lines connect the two states in a given session ($n = 28$ sessions, 12 mice). **i** The behavior of the mice differs in the two states. The "more engaged" state, the state with large pupil size and low neuronal synchrony, is associated with higher stimulus detection rates, as expressed by higher d-prime values. Error bars show mean ± SEM across sessions. **j** The psychometric function is steeper (left) and more sensitive (right) when the animal is engaged. Error bars show mean ± SEM across sessions, $n = 28$ sessions, 12 mice. The insets above the plots illustrate the measurement of psychometric curve width (left) and threshold (right).

increasing the activity of stimulus responsive neurons when the animal is most engaged in the task enhanced the detection of subthreshold stimuli, but suppressed the detection of suprathreshold stimuli (see also Supplementary Fig. 3). When the animal was less engaged (but still performing the task), we did not observe a reliable effect of cortical stimulation on behavior. Thus, behavioral state gates the effect of photostimulation, and photostimulation has a bidirectional effect depending on the contrast of coincident visual stimulation. Taken together, this suggests that primary visual cortex serves different roles depending on stimulus regime and the state of the animal.

## Network effects of targeted photostimulation

To explore the circuit mechanisms underlying the behavioral effects of photostimulation, we investigated the influence of photostimulation on downstream neurons. We used two-photon calcium imaging to simultaneously measure the activity in the local circuit during behavior with and without photostimulation in the two behavioral states. First, in the visual stimulus-only trials, we observed that increasing contrast increased population activity (Fig. 3a). The population average visually-evoked responses were larger in the more engaged state (Fig. 3b; average response magnitude in less-engaged vs more-engaged state: $0.029 ± 0.015$ vs $0.031 ± 0.018$, $P < 0.0001$ Wilcoxon signed-rank test; ref. 74), similar to the response modulation during locomotion[12,14]. These enhanced neural responses were strongest at 10% stimulus contrast and reflect the enhanced behavioral detection of the same stimuli (Fig. 1i). This neural enhancement in the engaged state held even when controlling for the different proportions of hit and miss trials (and thus the related motor or reward confounds) between the engagement states.

Does this enhanced excitability in the more engaged state underpin the gating of behavioral effect to photostimulation? We first analyzed how targeted photostimulation engages the local circuitry (Fig. 3c). We looked at the responses of either the directly targeted (Fig. 3d) or the non-targeted (background) subpopulations (Fig. 3e) on trials with and without photostimulation, across the different contrasts and behavioral states. As before, we observed increasing activity in both populations as the visual stimulus contrast increased, with more activity on average in the engaged state. When the target cells were

photostimulated, their activity was enhanced considerably as expected. In addition, a behavioral state dependence was observed: photostimulation was more effective in the engaged state, with larger photostimulation-evoked responses in the targeted cells (Fig. 3d; change in activity in target cells on less-engaged vs more-engaged trials: $0.68 ± 0.19$ vs $0.81 ± 0.22$, $P < 0.0001$ Wilcoxon signed-rank test). Conversely, the average activity of the non-targeted background cells was suppressed by photostimulation, but we observed no difference in the level of suppression between the states (Fig. 3e; activity in background cells on less-engaged vs more-engaged trials: $−0.005 ± 0.008$ vs $−0.005 ± 0.007$, $P = 0.231$ Wilcoxon signed-rank test) suggesting that the photostimulated cells recruit similar levels of local inhibition regardless of behavioral state. We investigated the effect of titrating the number of stimulated cells and observed that increasing the number of directly stimulated cells increased the amount of suppression of other cells in the network (Supplementary Fig. 5; ref. 34). In more engaged states the excitatory/inhibitory (E/I) balance is shifted towards excitation[81] consistent with the enhanced target cell response to direct photostimulation as observed in our data. However, the predominant effect of background suppression, with no difference between engagement states, confirms the dominance of inhibitory connections in cortical circuitry[82–89], revealing highly effective stabilization of network activity in either state, despite the E/I balance being shifted towards excitation.

To characterize the spatial spread of responses through the local network caused by photostimulation we constructed a photostimulation-triggered spatial average of the change in activity in all cells across all photostimulation trials aligned to the nearest target stimulation site. This analysis positions the directly targeted cells at the center of the map and the other cells in the local network relative to them (Fig. 3f). The activity-enhanced cells were localized in a narrow zone around each target site, corresponding to direct photostimulation, as well as potential synaptic recruitment of other cells (Supplementary Fig. 8). We observed a pronounced annulus of suppressed cells around the directly targeted cells (Fig. 3f) producing a center-surround motif of enhancement and suppression. No difference in spatial profile of the network influence of photostimulation was seen between behavioral states.

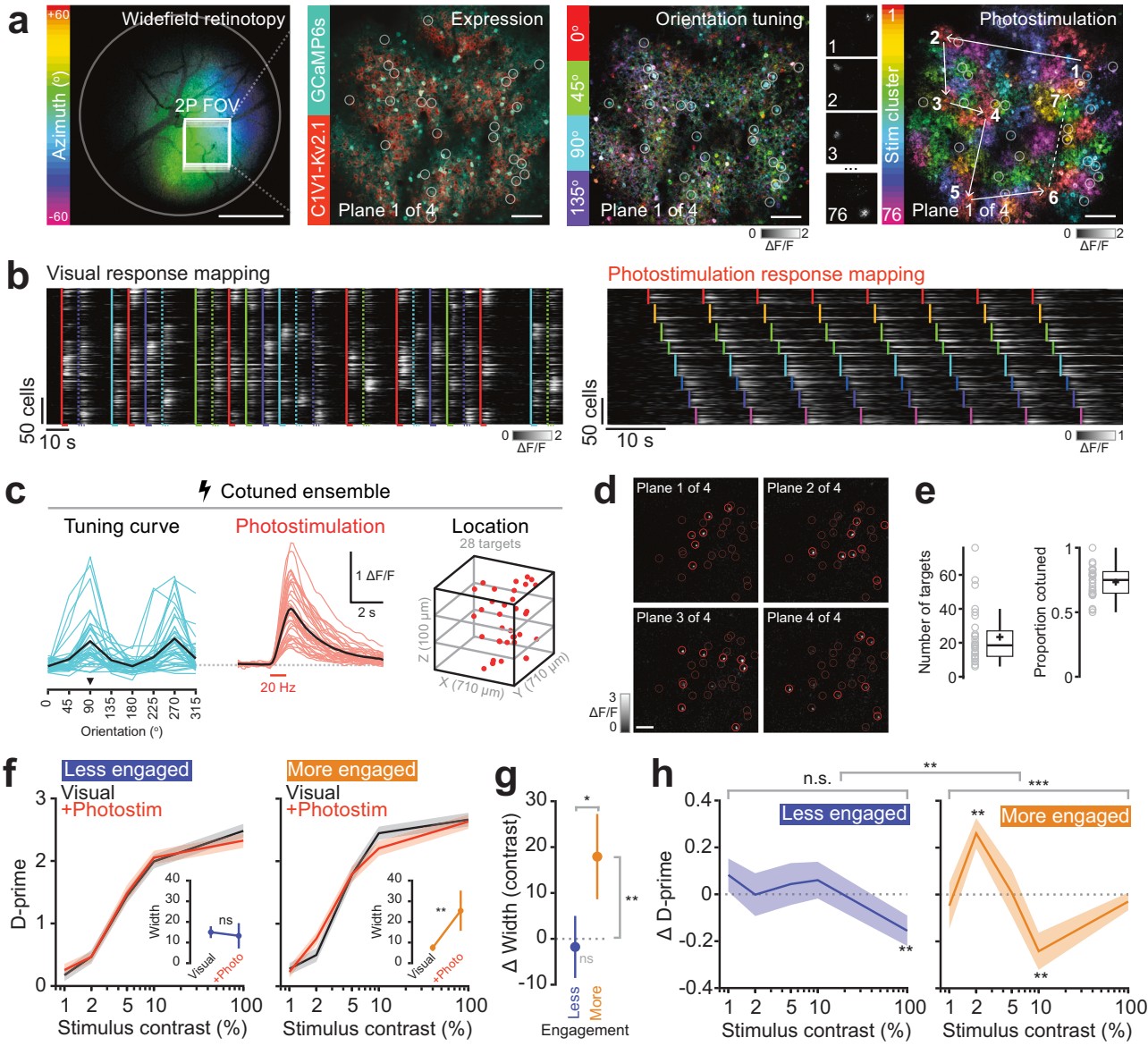

These findings suggest that the contrast-dependent behavioral effect of stimulating L2/3 V1 neurons (Fig. 2f–h) cannot be explained simply by the influence stimulated neurons have on the overall activity level of the local circuit, because the change in overall activity (pooled over target and background populations) across contrasts and states remains relatively constant (Supplementary Fig. 6).

**Functional similarity defines the network effects of targeted photostimulation**

It is known that functionally similar neurons are more highly interconnected[72,88,90,91], forming subnetworks which may facilitate computations to enhance the detection or discrimination of sensory stimuli[92]. To ask if different subnetworks of neurons respond differently across contrasts, we characterized the functional similarity of each recorded neuron relative to the photostimulated cell population. We defined similarity as the correlation of a neurons' contrast response curve with the average contrast response curve of all target neurons. This contrast response curve describes how neurons respond to the behaviorally relevant visual stimulus of increasing contrasts of a fixed drifting orientation (discrimination of orientation of the stimulus being behaviorally irrelevant in our task). The contrast-response curve of a given neuron is linked to the visual stimulus responsivity of each

cell and implicitly linked to the orientation preference (neurons preferring the displayed orientation will respond more strongly). Using the similarity of contrast curves between a background cell and the target neurons, we found that background cells which are more similar to target neurons in terms of their contrast response curves are also more similar to target neurons in terms of their visual response magnitude, trial-by-trial correlation and their orientation tuning curves (Supplementary Fig. 7A–D). We therefore refer to the contrast curve correlation between background and target cells, which captures intrinsic contrast responses as well as potential indirect network interactions, as our measure of behaviorally relevant functional similarity.

The functional similarity of background cells determines their response to target cell photostimulation. Using our measure of functional similarity we binned cells into groups of increasing similarity to the target population. We then plotted the response of each group as a function of visual contrast for visual-stimulation-only trials (Fig. 3g, left), for visual-and-photostimulation trials (Fig. 3g, middle) and the difference between the two (Fig. 3g right). Importantly, to cross-validate, we measured functional similarity using one half of all trials and measured visual and photostimulation responses using the other half. Sorting cells in this way again reveals increasing neural response

**Fig. 2 | Targeted photostimulation of stimulus coding neurons in L2/3 V1 bidirectionally modulates stimulus detection when mice are engaged in the task. a** From left to right, first panel: One-photon widefield imaging is performed while presenting drifting bars to the mouse to locate primary visual cortex. The two-photon FOV is positioned in a region with good GCaMP and opsin expression. The task visual stimuli are positioned in the retinotopically appropriate location. Scale bar represents 1 mm. Second panel: Example FOV (one plane from a 4-plane volume) showing construct expression in L2/3 mouse primary visual cortex. GCaMP6s is expressed transgenically and C1V1-Kv2.1 is expressed virally through injection. Third panel: Visual stimulus orientation preference map. 4 different orientations of drifting gratings are presented to the mouse. Pixel intensity is dictated by the stimulus triggered average response magnitude. Hue corresponds to preferred stimulus orientation. Fourth panel: Photostimulation responsivity of the FOV to clustered randomized photostimulation. The majority of recorded cells were grouped into 76 different clusters of 50 cells each (distributed across 4 planes) and targeted for sequential photostimulation to confirm responsivity prior to the experiment. Pixel intensity indicates the change in fluorescence caused by photostimulation. Color corresponds to the photostimulation cluster which caused the largest change in activity. White circles indicate example targets within this plane ultimately selected for targeted photostimulation of a co-tuned ensemble. All scale bars 100 μm. Examples shown are from one representative experiment but were repeated for each experiment in this study ($n = 28$ sessions, 12 mice). **b** Left: Example traces from the visual response mapping block of the experiment (prior to the behavioral task). Vertical lines indicate stimulus onsets with color indicating orientation (dashed lines for orientations over 135 degrees). Right: Example traces from the photostimulation response mapping block. Vertical lines indicate clusters of cells stimulated as a group. Cells are sorted by photostimulation cluster (in this experiment clusters were determined by orientation preference). **c** Example co-tuned stimulation ensemble. In this experiment 28 neurons were selected based on their responsivity to visual and optogenetic stimuli in **b**. Left: average responses to the drifting gratings of various orientations. Single lines represent individual neurons, thick black line indicates ensemble average. This ensemble shares a preference for 90 and 270 degree stimuli. Black triangle indicates the orientation of visual stimulus chosen for the remainder of this session. Middle: The ensemble response to photostimulation. Right: The spatial configuration of the neurons selected for stimulation. **d** Pixel-wise stimulus-triggered average (STA) showing the response of the FOV to the stimulation pattern from **c**. Red circles indicate target neurons on that imaging plane. Fade circles show all targets collapsed across planes. Scale bar represents 100 μm. **e** Boxplots showing the number and co-tuning of the photostimulated neurons across all experiments. Box shows the interquartile range (IQR), the plus sign shows the mean, solid line the median, and the whiskers denote 1.5 times the IQR ($n = 28$ sessions, 12 mice). **f** Photostimulation of co-tuned ensembles was paired with visual stimuli. Sessions are split into two states as in Fig. 1, here renamed to a 'less engaged' and a 'more engaged' state. Black lines indicate performance on visual-only trials. Red lines indicate performance on visual+photostimulation trials. Inset: Width of fitted psychometric curve. Error bars and shading show mean ± SEM across sessions, $n = 28$ sessions, 12 mice. Statistical test was two-sided Wilcoxon sign rank test, ** denotes $P < 0.01$. **g** The effect of photostimulation manifests as an increase in the width of fitted psychometric curves ($n = 28$ sessions, 12 mice). There was no change to the threshold of the psychometric functions. Error bars show mean ± SEM across sessions. Statistical tests were two-sided Wilcoxon sign rank, ** denotes $P < 0.01$, * denotes $P < 0.05$. **h** Photostimulation has a consistent effect on the detectability of visual stimuli in the 'more engaged' state and not in the 'less engaged' state. Photostimulation enhances detection of low (2%) contrast stimuli and suppressed the detection of higher (10%) contrast stimuli. Significance across each full curve indicates results of a one-way ANOVA within state. Significance across the two curves indicates the results of a repeated measures ANOVA. Significance above individual curve points indicates results of two-sided Wilcoxon signed-rank tests with Bonferroni (number of contrasts) correction. Error bars show mean ± SEM across sessions, $n = 28$ sessions, 12 mice. We use * to denote a $P$-value < 0.05, ** for $P < 0.01$ and *** for $P < 0.001$.

magnitude as the stimulus contrast increases (Fig. 3g, left), with cells functionally similar to the target cells responding positively to the visual stimuli and functionally dissimilar cells responding negatively. However, when looking at the change in activity caused by photostimulation – which was suppression on average across the whole background population – we observed a strong stimulus contrast dependence to the suppression of the most functionally similar neurons. Neurons that are most similar to the targeted neurons, which are more responsive to the behaviorally relevant visual stimulus, were increasingly more suppressed by photostimulation as visual contrast increased, whereas functionally dissimilar neurons were not (Fig. 3g right, h). As a result, in addition to a significant effect of contrast on the photostimulation response, there was also a statistically significant interaction between visual contrast and functional similarity (2-way ANOVA grouped by contrast and similarity: effect of contrast $F(4) = 11.9$, $P < 0.001$; effect of similarity $F(19) = 2.9$, $P < 0.001$; interaction of contrast and similarity $F(76) = 2.5$, $P < 0.001$). To summarize these effects, we next measured the slope of photostimulation induced change in activity as a function of contrast and plotted it against normalized similarity (Fig. 3i. Linear fit $R^2 = 0.94$, $P < 0.001$). This analysis showed that photostimulation induces a change in activity that depends systematically on how similar a neuron is to the targeted population. The more similar the responding neuron is to the targeted population, the more effective photostimulation is in suppressing it at high contrast. We observed similar contrast and functional-identity dependent patterns of influence when the similarity metric is contrast response similarity as above, orientation tuning curve similarity or the magnitude of visual responses (Supplementary Fig. 7E–J).

## Linking network activity to behavior

To relate the pattern of photostimulation-mediated changes in network activity (Fig. 3g, h) to the behavioral changes (Fig. 2h), we focused

on the threshold and threshold-adjacent contrast levels (reasoning that performance on the lowest contrast was unaffected by photostimulation because it is far below the detection limit, and that performance on the maximum contrast was unchanged because it is saturated). Since we showed that the effect of photostimulation on network activity depended on the functional identity of the cells in the network, we reasoned that any relationship between the network effects and behavioral effects of photostimulation must also depend on functional identity. We therefore examined each group of neurons characterized by their functional similarity individually (Fig. 4a, left). For example, we collected across all sessions the group of neurons most functionally similar to the targeted neurons and plotted the effect of photostimulation on their activity against the effect of photostimulation on behavior from those sessions (Fig. 4a, top right). We then measured the correlation of these network effects and behavioral effects, and refer to this measurement as the neural-behavioral coupling. We then repeated this analysis for each functional similarity group and plotted the neural-behavioral couplings as a function of similarity (Fig. 4a, bottom right; Fig. 4b, orange data). We performed the same analysis for data obtained in the disengaged state (Fig. 4b, blue data). We found that the neural-behavioral coupling of a given population of neurons was systematically related to the functional similarity of those neurons with the photostimulated neurons, with the coupling between neural and behavioral effects increasing as the functional similarity to the target neurons increases (mean correlation coefficient across resamples $r = 0.19$, $P < 0.001$ with respect to the shuffled distribution). Importantly, the same was not true of data obtained during the disengaged state (mean correlation coefficient across resamples $r = -0.04$, $P = 0.279$ with respect to the shuffled distribution). This indicates that the network effects of exogenous cortical activation can explain behavioral effects only when activation is performed during engaged states, and only when one interrogates the appropriate task-relevant neurons.

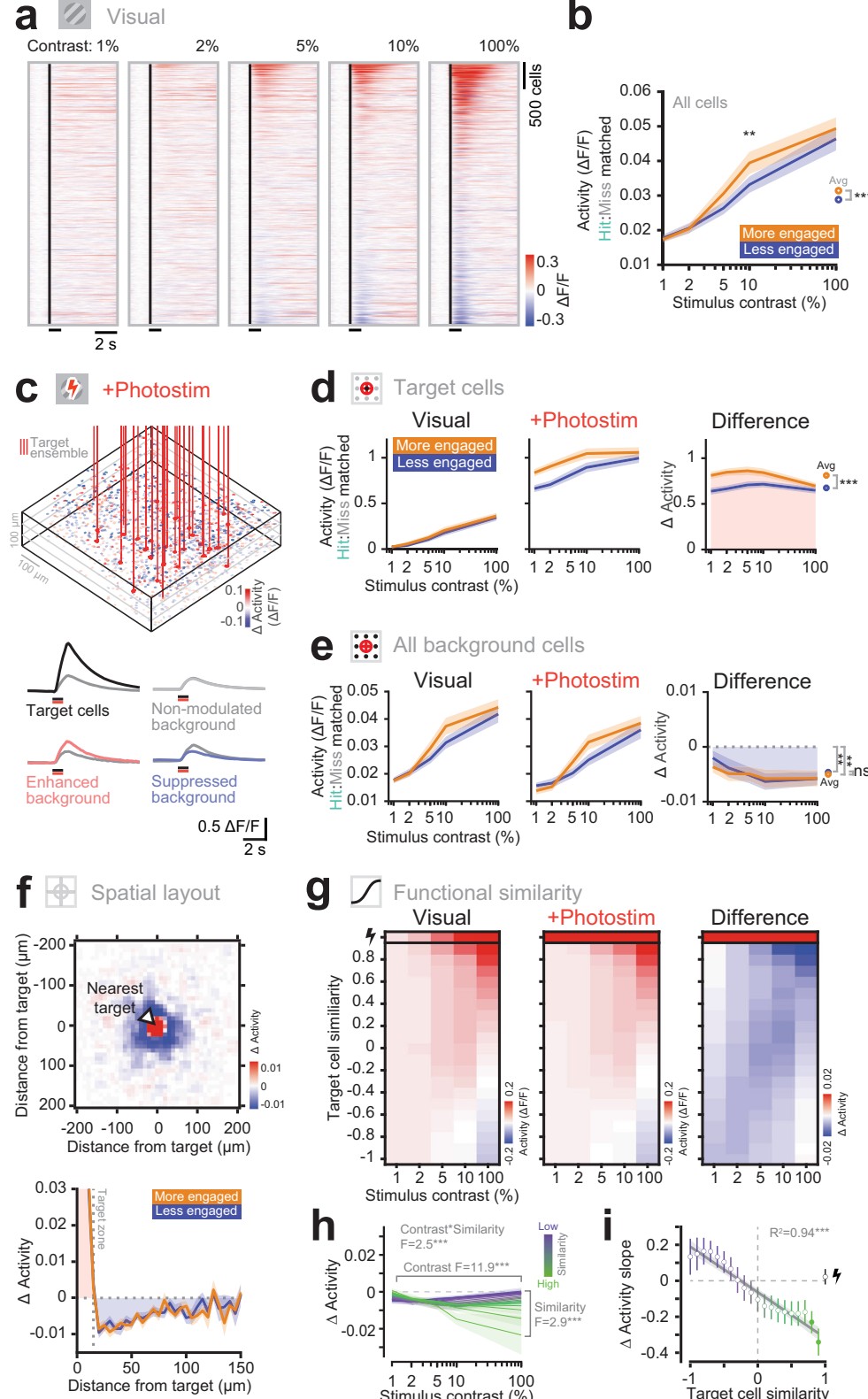

## Discussion

We demonstrate that the causal influence of augmenting task-related activity in L2/3 of mouse V1 depends on behavioral state: exogenous stimulation only translates into an effect on behavior when the animal is highly engaged in the task at hand. When the animal is less engaged, the additional cortical activity has no consistent influence, perhaps suggesting a different population of neurons are relied upon to solve the task in this state. We also show that the artificially generated patterns of cortical activity can either help or impair detection of the visual stimulus, depending on the strength of the visual stimulus the perturbation is delivered coincident with. In other words, the activity patterns evoked by photostimulation are altered by interaction with

**Fig. 3 | Targeted photostimulation elicits a suppressive network response that scales with the strength of coincident sensory stimuli and the functional identity of neurons. a** Example STA traces to visual only trials across contrasts in one experiment. Neurons are sorted by their average response across all contrasts. Black line indicates visual stimulus presentation. **b** The neural responses across the whole population in the more engaged state are significantly larger than in the less engaged state ($n = 28$ sessions, 12 mice). The ratio of hits and misses within each contrast have been matched across states. The line and shading represents the mean ± SEM. Individual points across the two curves were tested with two-sided Wilcoxon sign rank test with multiple comparison (Bonferroni) correction for number of contrasts. The average pooled responses were compared with a two-sided Wilcoxon sign rank test. *** denotes $P < 0.001$ and ** $P < 0.01$. **c** Top: example experimental volume. ROIs are colored by the change in activity caused by photostimulation during spontaneous gray screen periods. Vertical red lines indicate the directly stimulated cells. Bottom: example types of responses seen when the visual stimulus is paired with photostimulation. **d** Target cell responses. Left: responses to visual only trials across contrasts. Middle: responses to paired visual and photostimulation. Right: the change in activity caused by photostimulation. Target cells are strongly activated by photostimulation. The level of activation is significantly higher in the engaged state. The ratio of hits and misses within each contrast have been matched across states. The line and shading represents the mean ± SEM ($n = 28$ sessions, 12 mice). Comparisons were made with a two-sided Wilcoxon sign rank test. *** denotes $P < 0.001$. **e** Background cell responses. Left: The responses of all background neurons to visual only trials. Middle: The response of background neurons to paired visual and photo-stimulation. Right: The change in activity of background cells caused by photostimulation. The background cells are suppressed on average, across all contrasts. No difference is seen in the level of suppression between behavioral states. The line and shading represents the mean ± SEM ($n = 28$ sessions, 12 mice). Comparisons were made with a two-sided Wilcoxon sign rank test. ** denotes $P < 0.01$. **f** Top: the 2D spatial profile of photostimulation influence. All neurons are aligned relative to their nearest target spot (at 0,0), collapsed across z-planes. Spatially binned (5 ×5 μm), and Gaussian filtered

(SD = 10 μm) for display only. A central hotspot of activity corresponds to directly targeted neurons. Surrounding the targeted neurons the predominant effect of photostimulation is suppression of neighboring cells ($n = 28$ sessions, 12 mice). Bottom: the 1D spatial profile of photostimulation influence. No difference is seen between the two states. The line and shading represents the mean ± SEM ($n = 28$ sessions, 12 mice). **g** Presenting the average network responses across all sessions on two axes; visual stimulus contrast and similarity to the target cells (quantified as the Pearson's correlation of a given cell's contrast response curve to that of the average target neuron). The top row corresponds to the directly targeted neurons. Left: responses to visual stimuli of increasing contrast alone. Note cells positively correlated with the target cells respond positively to increasing contrast. Middle: The responses to paired visual and photo-stimulation. Right: The difference reveals the change in activity caused by photostimulation. Note that cells similar to the target cells are more strongly suppressed at higher contrasts ($n = 28$ sessions, 12 mice). **h** The change in activity of background cells. Background neurons most similar to the target neurons show the strongest levels of suppression mediated by the photostimulation (green line). The level of suppression recruited increases as the visual contrast increases. (2-way ANOVA grouped by state and contrast. Effect of contrast $F(4) = 11.9$, $P < 0.001$. Effect of similarity $F(19) = 2.9$, $P < 0.001$. Interaction of contrast and similarity $F(76) = 2.5$, $P < 0.001$, $n = 28$ sessions, 12 mice). Lines show the mean and shading shows the SEM. **i** The relationship between the slope of suppression recruited by increasing visual contrasts, versus the functional similarity to the photostimulated neurons. Neurons within a similarity group are pooled across experiments and the slope of suppression versus contrast is computed. Error bars indicate the standard error of the fit, obtained by resampling animals/sessions with replacement. Neurons most similar to the target neurons are increasingly suppressed as contrast increases, which corresponds to a negative slope. Filled individual points indicate individually significant fits with Bonferroni multiple comparison correction. The line of best fit indicates the relationship across functional similarity groups of all the background cells, shading represents the CI of the fit. The directly stimulated neurons are shown at the far right and are excluded from the fit ($n = 28$ sessions, 12 mice).

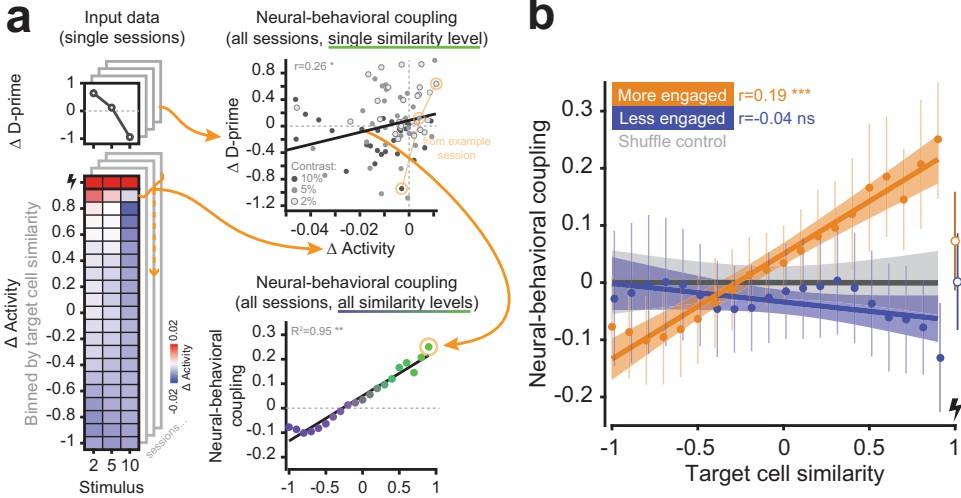

**Fig. 4 | The coupling between behavioral response and network response to photostimulation depends on the functional similarity of neurons, and is gated by brain state. a** Relating the photostimulation induced change in behavior to the photostimulation induced change in network activity within subpopulations of functionally defined neurons. An example schematic showing how the neural-behavioral coupling is computed. Left: example behavioral and neural data from an example session (only the intermediate contrasts (2, 5, 10%) are considered for the coupling analysis). Top right: All data across sessions belonging to one functional similarity group are used to compute the linear correlation coefficient between the change in behavior and the change in neural activity, termed the neural-behavioral coupling. Bottom right: We collect neural-behavioral coupling measurements for

all functional similarity groups of neurons. **b** The neural-behavioral coupling as a function of functional similarity (with respect to the target neurons). A relationship between the change in activity and the change in behavior is only seen in the more engaged state. We observe a monotonic relationship across functional similarity groups whereby the neurons most functionally similar to the stimulated targets show the tightest coupling with behavior. Separate data points at the far right are the directly targeted cells and not included in the fit. The error bars around individual datapoints are standard error (SE), the error bars around the fitted lines are SE, both are computed by resampling with replacement. Significance indicates the percentile of the shuffled distributed the real slopes lie in where *** refers to a $P$ value < 0.001. $n = 28$ sessions, 12 mice.

activity patterns evoked by sensory stimulation. These findings provide new insights into how cortex can have a flexible impact on sensory-guided behavior.

Why, and how, is the influence of cortical activity gated by behavioral state? In primates, turning attention to a particular location of visual space has two effects: enhanced sensory-evoked neural responses, and improved behavioral performance[93]. However, it is unclear whether the enhanced neural responses reflect or drive the perceptual improvement. Our results provide causal evidence that L2/3 of mouse V1 determines the perception of a stimulus only when mice are in an engaged state. When the mice are less engaged (but still performing the task) they may resort to a more reflexive, and perhaps sub-cortical, strategy. This gating may be implemented by relative weighting of the effect of visual cortical activity on higher cortical or sub-cortical areas, perhaps through neuromodulatory circuits[6]. One possible candidate is the superior colliculus (SC), which is involved in processing visual information and coordinating motor output[94,95]. The SC and visual cortex bidirectionally interact[96], with V1 projecting prominently to the SC[97], modulating visually evoked responses there[98] and behaviors controlled by it[99–102]. Indeed, inactivation of mouse SC impacts perceptual behavior[103]. Alternatively, gating may be implemented by differentially weighting activity arising from cortical or sub-cortical sources on a common downstream target, for example the pulvinar nuclei of the thalamus[104]. We observed task-related activity in cortex in the disengaged state, but our perturbation results suggest that this activity does not propagate as effectively or is not integrated as strongly downstream as in the engaged state. The more synchronized cortical oscillations in the less engaged state may impede the transmission of information between areas[105,106].

Interestingly, we did not observe a state-dependence of the local network response to photostimulation – the resulting suppression of activity was similar across the behavioral states, reflecting highly effective synaptic recruitment of local inhibition[83,84,86] which dominates the network response[89] despite the E/I balance being shifted towards excitation in more engaged states[81] (see also Fig. 3b). However, when the animal was more engaged in our task, neurons were activated more strongly by the visual stimulus (consistent with results from primates attending to a visual stimulus[107–109]), and the target cells themselves were more excited by the direct photostimulation. Therefore, in the task-engaged state, the photostimulated target cell activity and the activity of other cells that escaped local inhibition must be more effectively transmitted further downstream to unobserved areas[110]. Our results should inspire future work to investigate how activity propagates from L2/3 to the L5 output network and from there ultimately to other cortical, and sub-cortical, areas in different behavioral states.

What is the link between the network and the behavioral effects of photostimulation? In broad terms, as the behavioral effect of photostimulation became more suppressive, so did the network effects. While the behavioral effect at low contrast was facilitating, the local network effects were still negative on average, indicating that the strong direct drive of target cells outweighs the suppression of local or other unobserved downstream neurons. As the visual contrast increases the behavioral effect of photostimulation becomes negative, mirrored by more suppression of local neurons, suggesting that the relative contribution of the directly stimulated target cells is overridden by the greater levels of suppression in the network. Our finding that functionally similar neurons are more strongly affected by photostimulation as contrast increases, and that their activity is most strongly related to behavior, indicates that mapping the routing of information through functionally specific circuits is crucial to explaining behavior. Mechanistically, this effect may be mediated by functionally specific inhibitory connectivity[111]. Our core results on the network effects of targeted photostimulation (Fig. 3g–i) are consistent with a previously described like-suppresses-like motif[72]. Depending on

how we divide background cells into subpopulations we see varying degrees of facilitation of functionally similar cells at low contrasts (Supplementary Fig. 7E–J) as in other studies where predominantly excitatory network effects were observed[37,38]. Taken together our results align with predictions made by a recent modeling study[112] simulating the effect of optogenetic perturbations in a realistic model of visual cortex with a range of stimulus contrasts. In terms of the behavioral relevance of visual encoding, when the stimulus contrast is high, cortex may act to reduce redundancy through like-to-like suppression and ensure an efficient code[111,113–118]. Additional stimulation in this condition may go beyond reducing redundancy to suppressing behaviorally useful information, leading to a reduction in performance.

Our results are consistent with a dynamic allocation of cortical resources according to attention-like behavioral states[119] and thus highlight a modulatory role of cortex in a learned behavior likely also served by subcortical pathways. This perspective helps to reconcile apparently contradictory findings about the role of the cortex in behavioral tasks[42,43,49–51]. Our results are complementary to recent findings from lesion and silencing experiments[50,120] suggesting that some fully learned tasks are no longer cortically dependent, but go beyond those studies in showing that the influence of cortex can change even on very short timescales. Such short time-scale modulation by internal state is reminiscent of attentional processes, which may rely on mechanisms similar to what we study here, to modify the influence of cortex over behavioral output. This gating of cortical influence on behavior by attentional or behavioral states demonstrates that the causality of cortical activity depends on behavioral context and reminds us that no brain region acts in isolation[121–123]. Accounting for stimulation-induced suppression and state-dependent gating of downstream effects as described here will likely be important for the successful operation of future brain machine interfaces (BMIs).

## Methods
All experimental procedures were carried out under license from the UK Home Office in accordance with the UK Animals (Scientific Procedures) Act (1986).

### Animal preparation
We used transgenic GCaMP6s mice (Emx1-Cre;CaMKIIa-tTA;Ai94[69]) of both sexes aged between P49 and P67 (at time of surgery). Doxycycline treatment in drinking water from birth to P49 prevented interictal activity in the Ai94 mouse line[124]. Animals were kept on a 12 hr light/dark cycle at a temperature of 22 °C and 62% humidity. To prepare the mice for all-optical experiments, we first excised the scalp and implanted a metal headplate. We then removed the skull and dura overlying visual cortex, injected virus encoding the opsin and implanted a chronic cranial imaging window in place of the skull. Sterile procedures were used throughout. Before surgery, mice were given a subcutaneous injection of 0.3 mg/mL buprenorphine hydrochloride (Vetergesic) and anaesthetized with isoflurane (5% for induction, 1.5% for maintenance). The scalp above the dorsal surface of the skull was removed and an aluminum headplate with a 7 mm diameter circular imaging well was fixed to the skull centered over the right monocular primary visual cortex (2.5 mm lateral and 0.5 mm anterior from lambda) using dental cement. A 4 mm diameter craniotomy was drilled inside the well of the headplate, and the dura was then carefully removed. A calibrated pipette bevelled to a sharp point (inner diameter ~15 μm) connected to a hydraulic injection system (Harvard apparatus) was used to inject small volumes of virus (AAV2/9-CaMKII-C1V1(t/t)-mRuby2-Kv2.1). The virus (stock concentration: ~6.9 ×10^14 gc/ml) was diluted 10-fold in buffer solution (20 mM Tris, pH 8.0, 140 mM NaCl, 0.001% Pluronic F-68). We made ~5 insertions of the injection pipette, each site spaced by ~300 μm avoiding blood vessels. At each site we slowly lowered the pipette to a depth of 300 μm below

pia and injected 150 nl of the virus solution at 50 nl/min. After each injection the pipette was left in place for a further 3 min before slowly retracting. We then press-fit a chronic window (a 3 mm coverslip bonded to the underside of a 4 mm coverslip with UV-cured optical cement, NOR-61, Norland Optical Adhesive) into the craniotomy, sealed with cyanoacrylate (Vetbond) and fixed in place with dental cement (SuperBond). Following surgery, animals were monitored and allowed to recover for at least 7 days. After recovery we began behavioral training. All-optical experiments were then performed >3 weeks post-surgery, allowing for sufficient expression levels (animals were aged P77–P171, median = P122 at time of experiments).

## Behavioral training

We used an operant conditioning protocol whereby head-fixed mice were required to lick at a water spout positioned in front of them to report detection of a visual stimulus. Licks were recorded electrically. If the mice reported the presence of the stimulus correctly, a sugar water reward (10% w/v sucrose) was delivered through the water spout. The behavior hardware was controlled by custom software (PyBehaviour, https://github.com/llerussell/PyBehaviour) interfacing with an Arduino to trigger stimuli, record licks and deliver rewards. Mice had free access to food in their home cage, but access to water was limited to that acquired during the task. Mice had their weight monitored before and after daily training and were supplemented with additional water to maintain a minimum of 80% of their starting body weight. Before training, mice were habituated to handling and head restraint over 2 days. Training then took place in individual sound-dampened enclosures in which the mice were head-fixed and allowed to run on a treadmill. While not an integral part of the task design, we found that allowing mice to be free to run improved their performance in the task. Trials were triggered after mice withheld licks for $4 \pm 3$ s, after which a monocular visual stimulus appeared in the center of the monitor. If the mice licked at the water spout at any point within 1.5 s of the stimulus appearing a reward was delivered. In the first few days of training a reward was delivered automatically at 800 ms. Mice quickly learnt the requirements of the task and their reaction times preceded this automatic reward delivery time. After a few days the automatic reward delivery was disabled. After the stimulus and response window there was a fixed inter-trial period of 7 s before the next 'withhold' period was started. We also delivered randomly interleaved catch trials (no visual stimulus) to record chance rate of licking and assess accuracy in the task. Once stable performance was reached, we progressed the mice to a psychophysical variant of the task where we introduced a range of contrasts (1%, 2%, 5%, 10%, 100%) to assess their perceptual sensitivity and threshold. We found that task performance was insensitive to stimulus location on the monitor. For the final experiment the trial order was pseudo randomized to ensure that repeats of the same probe types were not immediately consecutive.

## Visual stimulation

Visual stimuli were generated using custom software (using PsychoPy, ref. 125). 30° Gabor patches of drifting sinusoidal gratings (8 directions, 0 to 315° in 45° increments) with a spatial frequency of 0.04 cycles per degree and a temporal frequency of 2 Hz were presented on a monitor (typically 51.8 cm width, 32.4 cm height, 15 cm from the animals left eye covering up to ±47° of the vertical visual field and ±60° of the horizontal visual field), with a spherical distortion applied to correct perspective errors.

**Training.** During training the orientation of the stimulus was randomized on every trial and the duration of the stimulus was 1 s. Rewards were delivered if the mouse licked during the response window regardless of the orientation of visual stimulus.

**Mapping orientation preference.** To map orientation preference of single cells in a FOV with two-photon imaging, the visual stimuli were positioned in the retinotopically appropriate location and were presented in a randomized order with a duration of 3 s, interleaved by 5 s of mean luminance gray. If mice licked at the water spout during this mapping block a water reward was delivered.

**Experiment.** During the behavioral experiments with photostimulation, the visual stimuli parameters were the same as during training, except only one orientation was presented (to match the photostimulation ensemble's preference) and the stimulus was positioned in the retinotopically appropriate location for the imaging field of view. A range of contrasts (1%, 2%, 5%, 10%, 100%) with equal trial proportions were presented to the animals.

## Widefield imaging

To locate primary visual cortex, and position the experimental field of view, widefield GCaMP imaging was performed (usable FOV-2 × 2 mm). GCaMP6s fluorescence produced by one-photon excitation (470 nm LED, Thorlabs) was collected through a 5x/0.1-NA air objective (Olympus) onto a CMOS camera (Hamamatsu ORCA Flash 4.0, binned image size of 512 × 512 pixels, 20 Hz frame rate). Contrast-reversing checkerboard bars, 10° wide were drifted vertically and horizontally across a gray screen at a speed of 25°/s in an interleaved sequence. Stimulus-triggered change in fluorescence for the two different stimuli revealed areal borders and identification of primary visual cortex[126]. This was repeated with two-photon imaging on the day of the experiment to confirm the retinotopic location of the chosen field of view.

## Two-photon population imaging

Two-photon imaging was performed with a resonant scanning microscope (Ultima II, Bruker Corporation) using a Chameleon Ultra II laser (Coherent) driven by PrairieView. A 16×/0.8-NA water-immersion objective (Nikon) was used for all experiments. An ETL (Optotune EL-10-30-TC, Gardasoft driver) was used to perform volumetric imaging, spanning a 100 μm range with 33.3 μm spacing between 4 planes. The FOV size was 710 × 710 μm at a resolution of 512 × 512 pixels. The number of cells recorded (ROIs after curation) per experiment ranged from 1266 to 4891 (mean = 2765 ± 995). The per-plane frame rate was 7 Hz (total acquisition rate 30 Hz). GCaMP6s was imaged at 920 nm and mRuby (conjugated to C1V1-Kv2.1) was imaged at 765 nm. Power on the sample was 50 mW at the shallowest plane (-150–200 μm below pia) and increased to ~85 mW at the deepest plane (-250–300 μm below pia), interpolating for intermediate planes, to equalize imaging quality across planes. To maximize imaging quality[127] we calculated the tilt of the sample relative to the microscope and then rotated the objective along two axes to be perpendicular to the implanted coverslip window.

## Two-photon photostimulation

Two-photon photostimulation was carried out using a fiber laser at 1030 nm (Satsuma, Amplitude Systèmes, 2 MHz repetition rate). The laser beam was split via a reflective spatial light modulator (SLM) (7.68 × 7.68 mm active area, 512 × 512 pixels, OverDrive Plus SLM, Meadowlark Optics/Boulder Nonlinear Systems) which was installed in-line of the photostimulation path (NeuraLight 3D, Bruker Corporation). Phase masks used to generate focused beamlet patterns in the sample were calculated via the weighted Gerchberg-Saxton algorithm. The targets were weighted according to their location relative to the center of the SLM's addressable FOV to compensate for the decrease in diffraction efficiency when directing beamlets to peripheral positions. We calibrated the targeting of SLM spots in imaging space by burning arbitrary patterns with the SLM using the photostimulation laser in a fluorescent plastic slide before taking a volumetric stack of the sample

with the imaging laser. We manually located the burnt spots and the corresponding affine transformation from SLM space to imaging space was computed. For 3D stimulation patterns we interpolated the transformation required from the nearest calibrated planes (Calibration code: https://github.com/llerussell/SLMTransformMaker3D). To increase stimulation efficiency, we offset the photostimulation FOV with the photostimulation galvanometers such that the center of SLM space was close to the cortical/imaging-space centroid of targeted cells. Spiral photostimulation patterns (3 rotations, 10 μm diameter, 20 ms duration) were generated by moving all beamlets simultaneously with the galvanometer mirrors. The laser power was adjusted to maintain 6 mW per target cell. Photostimulation during the behavioral task was delivered at 20 Hz (1 spiral every 50 ms) for 1 s with the same onset time as the visual stimulus.

## Characterization of photostimulation resolution with pharmacological blockade
To characterize the stimulation resolution, we stimulated single cells in separate non-behavioral sessions with the same stimulation parameters as used in the experiments (15 μm spiral, 20 ms spiral duration, repeated every 50 ms (20 Hz) for 1 s). We offset the stimulation spot using the SLM by 0, 5, 10, 15, 20, 30 μm laterally and −75, −50, −25, −10, 0, +10, +25, +50, +75 μm axially (at 0 μm lateral offset). In each animal (n = 3 mice), we stimulated 8 single cells: one at a time every 2 s, for 10 repeats of each stimulation offset for each cell. In these experiments, we used animals implanted with chronic windows with a small access hole drilled in the middle, covered with a silicone plug[128]. By removing the plug we gained access to the brain surface (dura removed prior to window implantation), to which we could apply pharmacological agents. We applied a mixture of 1 mM NBQX and 2 mM AP5 (in IVE)[129,130] to block excitatory synaptic activity to disambiguate off-target stimulation from synaptic recruitment of nearby cells.

## Naparm (Near automatic photoactivation response mapping)
To find photostimulation-responsive cells we semi-automatically detected cell locations from expression images and stimulus-triggered average or pixel-correlation images (STA Movie Maker, https://github.com/llerussell/STAMovieMaker) fed into Cellpose[131] and manually curated. These cell body coordinates were then clustered into equal size groups of user-determined size (between 10 and 50) and the groups were stimulated one by one. The associated phase mask, galvanometer positioning, and Pockels cell control protocol were generated with custom MATLAB software (Naparm, https://github.com/llerussell/Naparm) and executed by the photostimulation modules of the microscope software (PrairieView, Bruker Corporation) and the SLM control software (Blink, Meadowlark). For photo-responsivity mapping purposes, we used a stimulation rate of 20 Hz, for 500 ms per pattern, stimulating a different pattern every 1.5 s, and performed 8–10 trials of each pattern. These data were then analyzed online together with the visual response mapping data to extract activity traces and design stimulation ensembles (see below).

## Synchronization
For subsequent synchronization during analysis, analog signals of various trigger lines were recorded with a National Instruments DAQ card, controlled by PackIO[132]. The recorded inputs included two-photon imaging frame pulses, photostimulation triggers, galvanometer command signals, triggers to and frame flip pulses from the visual stimulus and the SLM phase mask update. Photostimulation trials for the responsivity mapping block were triggered at a fixed rate from an output line on the DAQ card. For the online behavior experiments photostimulation and visual trials were triggered through the behavior software and hardware.

## Experimental protocol
On the day of the full experiment, the following protocol was used. First, we located a region of cortex showing optimal coexpression of opsin and indicator, guided by widefield retinotopy, and confirmed the corresponding retinotopic location with two-photon imaging. After determining where to position the visual stimulus on the monitor we then presented drifting gratings of 8 different orientations while performing two-photon imaging to map orientation preferences of the recorded cells. Rewards were delivered during the visual stimuli if the mouse licked. Next, we photostimulated a large proportion of all cells in the recorded volume to find which ones were photostimulation-responsive. Finally, we designed photostimulation patterns for use in the behavior experiment (see below). We then gave the mice ~10 warm up 'easy' (high contrast) trials before the main behavioral experiment began. We recorded in 20 min blocks, manually correcting for any drift in imaging FOV between recordings.

## Online photostimulation ensemble design
To increase the speed of data analysis immediately prior to the experiment, we streamed the raw acquisition samples to custom software (PrairieLink, RawDataStream, https://github.com/llerussell/Bruker_PrairieLink). We used this raw stream to process the pixel samples, construct imaging frames, and perform online motion correction. Processing online allowed us to directly output to a custom file format making the data immediately available for analysis. Motion-corrected movies were loaded into MATLAB (MathWorks), and traces were extracted from both the photostimulation and the visual stimulation movies, using the photostimulation targets as seed points around which circular ROIs were dilated. We subtracted a neuropil signal from the ROI signal before determining responsivity. We determined cells as photostimulation-responsive if their evoked response (in a-500 ms window after stimulus offset) to their direct stimulation was >30% ΔF/F on >50% trials. We determined cells as visually responsive by the same criteria (with response window of 2 s during the stimulus presentation), additionally specifying their preferred orientation as the stimulus that elicited the largest average response. The final stimulation pattern was then designed in each experiment. After filtering for photostimulation-responsive cells, the co-tuned group was selected, taking the largest group of photostimulation-responsive and orientation-tuned neurons (minimum number of targets: 6, maximum: 73, median: 23).

## Pre-processing: Imaging frame registration, ROI segmentation and neuropil correction
For the final analysis, the raw calcium imaging movies were pre-processed using Suite2p[133]. The pipeline included image registration, segmentation of active region of interest (ROIs), and of local surrounding neuropil signal. The final selection of ROIs was filtered semi-automatically using anatomical criteria to include only neuronal somata and discard spurious ROIs. We manually inspected all FOVs to ensure consistent results. We subtracted a neuropil signal from every ROI signal. The contamination of the ROI signal by the neuropil signal depends on many factors, including expression levels, imaging quality, and axial sectioning by the imaging plane. We used robust linear regression to estimate the coefficient of neuropil contamination for each ROI (Supplementary Fig. 9; ref. 68). The slope of this fit was used to scale the neuropil signal before subtraction from the ROI signal, such that after subtraction there was no correlation between the ROI baseline and neuropil. Neuropil subtraction had minimal effect on the response magnitude, and negative responses to visual- and photo-stimulation were seen even without subtracting the neuropil contamination (Supplementary Figs. 9 and 10).

### ROI exclusion zones

To reduce potential off-target photostimulation artifacts, we excluded from consideration all cells within a 30 μm diameter cylinder extending through all axial planes when analyzing the network response to photostimulation due to potential imaging and photostimulation artifacts (see Supplementary Fig. 9). We redefined our target stimulation pattern identities based on the ROIs segmented by Suite2p within the 30 μm lateral disk around each of the SLM target locations. We also excluded ROIs in the first 100 rows of pixels of each imaging frame due to an ETL artefact related to the settle time of the lens when changing planes.

### Neuronal response metric

To measure neuronal responses, we extracted the mean fluorescence in a ~500 ms window (4 frames) starting immediately after the photostimulation ended (and/or visual stimulus to ensure comparable measurements) and subtracted the mean fluorescence in the ~1 s baseline (7 frames) before the onset of photostimulation (or visual stimulus). We divided the difference in the means by the mean of the baseline window, resulting in trial-by-trial ΔF/F values. We excluded all photostimulation frames because of the associated artefact contaminating the activity traces; the slow kinetics of GCaMP6s permit this, although the magnitude of response is underestimated as a result.

### Hit:miss ratio matching

To counteract motor and reward-related confounds of licks occurring within the neural analysis window, we ensured that the ratio of hits to misses (trials with licks vs trials with no licks) were equalized across comparisons. Ensuring a 50:50 ratio of hits to misses across all trial types in a given session would result in unnecessary loss of data in trials where hits or misses made up the entirety of trial responses. The more relevant control to make is not across contrasts, but across behavioral states (the more-engaged state versus the less-engaged state) or trial type (with or without photostimulation). We therefore ensured that the proportion of hits to misses at a particular stimulus contrast were equal across a given comparison (state or photostimulation). We determined the minimum number of hits on a given contrast across the conditions and the minimum number of misses on a given contrast across conditions (e.g., at 5% contrast in both states, or at 5% contrast in the less-engaged state with and without photostimulation). We then resampled without replacement taking the minimum common number of hits and the minimum number of misses from the original trials. As the minimum number is computed across conditions, the resampled collections of trials have the same proportion of hits to misses in both conditions. We then averaged the resampled trials together and repeated this procedure 10,000 times, storing the average result for a given neurons response in that condition.

### Contrast response similarity metric

To functionally characterize all recorded neurons relative to the photostimulated neurons, we compared their responses to different contrasts of the visual stimulus (only one orientation was used) presented in the final behavioral experiments. As our functional similarity measurement (correlation of average responses to visual stimuli) and photostimulation mediated change metric (average difference in responses between trials with visual stimuli and photostimulation and trials with just visual stimuli) would share the same datapoints we cross-validated. Specifically, we randomly split the dataset in half and used one half for the contrast similarity measurement and the other half for the measurement of photostimulation induced changes. For the contrast similarity metric in a given session we averaged together the photostimulated population to give a group average response curve. We then computed the Pearson's correlation of all single cell contrast response curves with the photostimulated population average curve. For the photostimulation mediated change in activity of a given cell we averaged together responses in all trials of a particular contrast and stimulation condition. We subtracted the visual only trial responses from the joint visual and photostimulation trial responses. We binned all neurons per session into 20 evenly spaced bins ranging from maximally dissimilar to maximally similar to the target group. We computed the average response within each similarity group. We performed all analyses in Figs. 3h, i and 4 with these cross-validated metrics, repeating the procedure 5000 times with different random trial splits, computing the metric of interest within each split and reporting the average across splits.

In Supplementary Fig. 7 we also characterize neurons based on their orientation tuning curves (8 different orientations were presented in a short mapping session before the main photostimulation experiment) and computed the Pearson's correlation of those single cell tuning curves to the photostimulated population average tuning curve.

### Neural-behavioral coupling

To obtain bootstrapped statistics of the neural-behavioral coupling we performed the following shuffling and resampling procedures. We split the network into functional similarity bins and compute the cross-validated change in activity (see above). For each similarity bin we correlate the average response within group, and pooled across sessions, to the 3 intermediate contrasts (2, 5, 10%) with the change in d-prime on the same contrasts. Each similarity group from each session thus contributes 3 datapoints to the neural-behavioral coupling measurements. This correlation coefficient is termed the neural-behavioral coupling. We then performed a linear regression on the neural-behavioral couplings as a function of functional similarity to summarize the overall effect whereby most highly similar neurons have the strongest coupling with behavior. We bootstrapped the neural-behavioral coupling fit through the functional similarity groups by resampling sessions with replacement, 5000 times. We report the mean slope obtained through this resampling procedure. To ask whether the resampled distribution of slopes is different from those expected by chance given the dataset we performed a shuffling procedure to derive a null distribution against which to compare. For this, we shuffled trial contrast identity amongst the intermediate contrasts, mixing behavioral and neural responses across trials but maintaining the overall statistics within sessions. We repeated this shuffle 5000 times and within each shuffle we computed cross-validated contrast similarity and photostimulation response metrics. We averaged these responses across shuffle to obtain the final shuffled behavioral and neural responses. With these shuffled responses we performed the same neural-behavioral coupling procedure to generate a null distribution of slopes expected by chance.

### Behavioral session truncation

To ensure we only analyzed periods of the behavioral session where the mice were similarly engaged and motivated, we truncated the session when the rolling average performance (20 trial sliding window) of the 'easy' highest contrast trials dropped below 80% of the starting performance.

### Data exclusion criteria

We excluded trials if >50% of photostimulation targets failed to respond on that trial. We also excluded trials if the mice licked early (within the first 150 ms of the presentation of the visual stimulus). Whole sessions were excluded if fewer than 10 trials in any trial type remained (the median minimum number of trials per trial type (note each session has 12 trial types) in included sessions = 31 trials (range 10–56)). Out of 32 completed sessions, 3 were excluded because of poor photostimulation efficiency.

## Statistical procedures

No statistical methods were used to predetermine sample size. Investigators were not blinded to allocation during experiments and outcome assessment. Summary statistics in the text are reported as mean ± SD unless otherwise indicated. Solid lines and shaded areas or error bars in plots represent mean ± SEM unless otherwise indicated. Statistical tests used are specified in the text and were generally paired, two-tailed and non-parametric. We use the following convention for representing $P$-values in all figures: $P >= 0.05$ (n.s.), $P < 0.05$ (*), $P < 0.01$ (**), $P < 0.001$ (***).

## Pre-trial pupil size

We recorded the right pupil throughout the experiment using a Dalsa Genie Nano-M1280 camera with a Kowa LM50JC lens. The camera frames were acquired at 30 Hz triggered by pulses from the 2P imaging system. The laser light used for 2P imaging, evident through the back of the pupil, was blocked from the camera using a 900 nm shortpass filter. The animal was illuminated with an 850 nm LED. The visual stimulus monitor provided ambient light. We used DeepLabCut[134] to track 11 points (6 around the circumference of the pupil, 4 around the eye lid, 1 on the nose and 1 on the tongue (only visible when the animal licked), the tongue point allowed for synchronization with the imaging and behavioral data by cross-correlating the DeepLabCut tongue signal with the electrically recorded lickometer signal. The area bounded by the 6 pupil points was used as the pupil size. We normalized the pupil size by the median size across the whole session.

## Pre-trial synchrony

To compute the network synchrony prior to presentation of the visual stimulus, we used deconvolved activity traces (OASIS[133,135]) smoothed with a Gaussian filter (sigma = 0.5 s). We used a 4 s window immediately prior to the initiation of the trial (delivery of a stimulus, if not a catch trial) as the 'pre-trial' period. We then computed pairwise Pearson's correlations between all cells within this time window and averaged together all pairwise correlation coefficients across all cells (including targets) to give the total network correlation or synchrony. We z-scored all network correlations within animal and across all trial types to facilitate across animal comparisons. When comparing hit and miss trials, we resampled 10,000 times to match trial numbers.

## Pre-trial state

We used both the pupil-size and neural synchrony to produce a combined measure of state trial-by-trial. To compute this, we z-scored both variables and summed them, weighting the pupil by −1 to account for the inverse relationship between the two variables. Based on this combined state score we median split each session into two equal size subsets, corresponding to two behavioral states.

## Psychometric curve fitting

We used Psignifit[136] to fit a Weibull curve fixing the lambda (lapse rate) and gamma (guess rate) parameters, while allowing the estimation of alpha and beta (threshold and slope). The threshold of the curve is defined as the stimulus contrast where behavioral performance is 50%. The width of the curve is defined as the difference in stimulus units between 5% and 95% behavioral performance.

We computed d-prime for each stimulus type to control for chance rate of licking within a session using the catch trials without photostimulation as the false alarm rate. We also observe the same results when computing d-prime using the catch trials with photostimulation as the false alarm rate (Supplementary Fig. 4).

## Reporting summary

Further information on research design is available in the Nature Portfolio Reporting Summary linked to this article.

## Data availability

Datasets supporting the findings of this study are available from the corresponding author upon request. Source data are provided with this paper.

## Code availability

Custom code used for data acquisition, photostimulation control, behavioral training and analysis have been deposited online: Naparm https://github.com/llerussell/Naparm, https://doi.org/10.5281/zenodo.10449686, PyBehaviour https://github.com/llerussell/PyBehaviour, https://doi.org/10.5281/zenodo.10449684, 3D SLM calibration https://github.com/llerussell/SLMTransformMaker3D, https://doi.org/10.5281/zenodo.10449682, STAMovieMaker https://github.com/llerussell/STAMovieMaker, https://doi.org/10.5281/zenodo.10449680, RawDataStream https://github.com/llerussell/Bruker_PrairieLink, https://doi.org/10.5281/zenodo.10449690, Objective rotation https://github.com/llerussell/MONPangle, https://doi.org/10.5281/zenodo.10449688.

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

## Acknowledgements

We thank Matteo Carandini, Beverley Clark, Kenneth Harris, Nick Steinmetz and Arnd Roth for helpful discussions and comments on the manuscript; Brendan Bicknell, Dustin Herrmann and Evelyn Wong for comments on the manuscript; Soyon Chun and Agnieszka Jucht for mouse breeding; Maite Marcantoni and Florence Bui for behavioral training during pilot experiments; Arthur Gretton and Peter Latham for analysis advice; and Bruker Corporation for technical support. This work was supported by grants from the Wellcome Trust (PRF 201225 and 224688), ERC (AdG 695709), MRC (MR/T022922/1) and the BBSRC (BB/N009835/1).

## Author contributions

L.E.R. performed surgeries, trained animals, performed experiments, and analyzed data. Z.Y. and L.P.T. trained animals. M.F. and A.M.P. provided valuable advice about experimental design and analysis. L.E.R. built hardware and software for control of behavioral training and for calibration and control of photostimulation. H.W.P.D. helped build photostimulation control software. S.C. and C.D.H. provided C1V1-Kv2.1 virus. L.E.R. and M.H. conceived and designed the study and wrote the manuscript with input from all authors.

## Competing interests

The authors declare no competing interests.
