## [Peer Review File · Nature Communications]

REVIEWERS' COMMENTS

Reviewer #1 (Remarks to the Author):

In this work Russell et al. stimulate patterns of V1 neurons as mice detect the onset of visual stimuli. They find differences in the sign of the behavioral effects as a function of visual stimulus contrast.

This work is a solid contribution in its present state. I have reviewed prior versions of the manuscript and this submission is improved. There is now a clear discussion of why neural and behavioral effects of stimulation differ. I am not sure that the only explanation is that downstream neurons in unobserved areas process these signals differently, but it is certainly possible.

Thank you for the nicely-written manuscript.

Reviewer #2 (Remarks to the Author):

In this revision, the authors have substantially increased the clarity, rigor and richness of their results. I am particularly intrigued by the analyses dividing the non-target cells according to the similarity of their contrast response functions with the target cells. This analysis has surprising explanatory power for both the modulation of neurons by stimulation as well as the correlation with behavior. It also matches well with a developing literature on the complex competition that occurs between neurons that depend on their stimulus preferences. I only have a few suggestions that I hope will support the readers' understanding of this exciting work.

1. I'd first like to respond to the other reviewers' lack of surprise at the behavioral-state dependence of the results, as I find this result truly unexpected (almost to the point of disbelief). Notably, while the authors have divided behavioral trials according to engagement as defined by pupil size and neural synchrony, the differences in behavioral performance are actually relatively subtle. There is only a 20%

decrease in d' , 20% increase in threshold, almost no change in lapse rate and no change in FA rate. Thus, as the authors aptly describe it, they are looking at “more” vs “less” engaged, not “engaged” vs “disengaged”. Indeed, my expectations would match the other reviewers’ if we were looking at a disengaged state- I would imagine that this gating is happening somewhere downstream of V1, and that stimulation of V1 would not reliably impact behavior (whatever that might be) under this condition. Yet, that the behavioral performance could appear to be nearly identical while the mice are relying on a different population of neurons or brain area is truly surprising to me. Perhaps the authors could do a better job pointing out the relative subtlety in the changes to behavior with state, and therefore impress upon readers the unexpected nature of this finding.

2. The authors argue in the Results (p6, paragraph 3) that “the behavioral effect of stimulating L2/3 V1 neurons cannot be explained simply by the influence stimulated neurons have on the overall activity of the local circuit because the change in overall activity across contrasts and states remains relatively constant”. This is not entirely true, as there is a smaller effect of photostimulation at high contrasts. They then go on to argue in the Discussion (p10, paragraph 1) that the facilitation at low contrasts can’t be due to the influence of the background neurons because they are on average suppressed, and therefore the downstream decoder must be selectively detecting the activated population. It seems like a more parsimonious conclusion is that the downstream decoder is integrating all neurons, and that at low contrasts the strong direct drive of the target cells (and comparatively weak suppression from the background cells) improves detection, whereas the relative contribution shifts at higher contrasts.

3. There remains a statement in the Results (p5, paragraph 2) that the cortex serves a different role according to “task demands” - however, this should be stimulus regime, not “task demands”. In addition, it’s not clear that the cortex serves a different role across stimulus regime, instead it responds differentially to stimulation as a function of contrast.

Response to Reviewers

Reviewer 1

In this work Russell et al. stimulate patterns of V1 neurons as mice detect the onset of visual stimuli. They find differences in the sign of the behavioral effects as a function of visual stimulus contrast.

This work is a solid contribution in its present state. I have reviewed prior versions of the manuscript and this submission is improved. There is now a clear discussion of why neural and behavioral effects of stimulation differ. I am not sure that the only explanation is that downstream neurons in unobserved areas process these signals differently, but it is certainly possible.

Thank you for the nicely-written manuscript.

We are grateful to the reviewer for their help in improving the manuscript. No changes are necessary based on their final comments.

Reviewer 2

In this revision, the authors have substantially increased the clarity, rigor and richness of their results. I am particularly intrigued by the analyses dividing the non-target cells according to the similarity of their contrast response functions with the target cells. This analysis has surprising explanatory power for both the modulation of neurons by stimulation as well as the correlation with behavior. It also matches well with a developing literature on the complex competition that occurs between neurons that depend on their stimulus preferences. I only have a few suggestions that I hope will support the readers' understanding of this exciting work.

1. I'd first like to respond to the other reviewers' lack of surprise at the behavioral-state dependence of the results, as I find this result truly unexpected (almost to the point of disbelief). Notably, while the authors have divided behavioral trials according to engagement as defined by pupil size and neural synchrony, the differences in behavioral performance are actually relatively subtle. There is only a 20% decrease in d' , 20% increase in threshold, almost no change in lapse rate and no change in FA rate. Thus, as the authors aptly describe it, they are looking at "more" vs "less" engaged, not "engaged" vs "disengaged". Indeed, my expectations would match the other reviewers' if we were looking at a disengaged state- I would imagine that this gating is happening somewhere downstream of V1, and that stimulation of V1 would not reliably impact behavior (whatever that might be) under this condition. Yet, that the behavioral performance could appear to be nearly identical while the mice are relying on a different population of neurons or brain area is truly surprising to me. Perhaps the authors could do a better job pointing out the relative subtlety in the changes to behavior with state, and therefore impress upon readers the unexpected nature of this finding.

2. The authors argue in the Results (p6, paragraph 3) that “the behavioral effect of stimulating L2/3 V1 neurons cannot be explained simply by the influence stimulated neurons have on the overall activity of the local circuit because the change in overall activity across contrasts and states remains relatively constant”. This is not entirely true, as there is a smaller effect of photostimulation at high contrasts. They then go on to argue in the Discussion (p10, paragraph 1) that the facilitation at low contrasts can’t be due to the influence of the background neurons because they are on average suppressed, and therefore the downstream decoder must be selectively detecting the activated population. It seems like a more parsimonious conclusion is that the downstream decoder is integrating all neurons, and that at low contrasts the strong direct drive of the target cells (and comparatively weak suppression from the background cells) improves detection, whereas the relative contribution shifts at higher contrasts.

3. There remains a statement in the Results (p5, paragraph 2) that the cortex serves a different role according to “task demands”- however, this should be stimulus regime, not “task demands”. In addition, it’s not clear that the cortex serves a different role across stimulus regime, instead it responds differentially to stimulation as a function of contrast.

We thank the reviewer for their help in improving the manuscript and especially their support in appreciating the novelty of the results. We have made a number of changes to the text to address their remaining comments – which we think have helped to highlight the significance of our work and improve ease of understanding.

- We have rewritten the abstract (to reduce word count) and now make explicit reference to the state difference.
- We have added a section to the end of the Introduction and to the first Results section highlighting the subtle difference in behavioral state.
- We have rewritten the sentences in the Discussion on the downstream decoder. We agree that the previous version was unclear, and the interpretation was meant as the reviewer interpreted it.
- We have removed reference to “task demands”.